# JcvPCA and JsvCRP: A set of metrics to evaluate changes in joint coordination strategies

Océane Dubois[1]*, Agnès Roby-Brami[2], Ross Parry[2], Nathanaël Jarrassé[1]

**1** Sorbonne Université, CNRS, INSERM, Institute for Intelligent Systems and Robotics (ISIR), Paris, France, **2** LINP2, UPL, UFR STAPS, Université Paris Nanterre, 200 Avenue de la République, 92001 Nanterre, France

\* dubois@isir.upmc.fr

**Data availability statement:** All relevant data are within the manuscript and its Supporting Information files.

## Abstract

Characterizing changes in inter-joint coordination presents significant challenges, as it necessitates the examination of relationships between multiple degrees of freedom during movements and their temporal evolution. Existing metrics are inadequate in providing physiologically coherent results that document both the temporal and spatial aspects of inter-joint coordination. In this article, we introduce two novel metrics to enhance the analysis of inter-joint coordination. The first metric, Joint Contribution Variation based on Principal Component Analysis (JcvPCA), evaluates the variation in each joint's contribution during series of movements. The second metric, Joint Synchronization Variation based on Continuous Relative Phase (JsvCRP), measures the variation in temporal synchronization among joints between two movement datasets. We begin by presenting each metric and explaining their derivation. We then demonstrate the application of these metrics using simulated and experimental datasets involving identical movement tasks performed with distinct coordination strategies. The results show that these metrics can successfully differentiate between unique coordination strategies, providing meaningful insights into joint collaboration during movement. These metrics hold significant potential for fields such as ergonomics and clinical rehabilitation, where a precise understanding of the evolution of inter-joint coordination strategies is crucial. Potential applications include evaluating the effects of upper limb exoskeletons in industrial settings or monitoring the progress of patients undergoing neurological rehabilitation.

## 1 Introduction

Inter-joint coordination refers to the dynamic relationships between joint movements during motion. Understanding these relationships is crucial in various fields, including movement science, neurology, and biomechanics. Changes in inter-joint coordination can be indicative of motor learning, pathology progression, or adaptation to external factors such as assistive devices [1,2]. For instance, analyzing joint coordination can provide insights into children's motor development [3], enhance sports performance by refining movement patterns [4,5],

**Funding:** ANR EXOMAN (ANR-19-CE33-0009). The funders had no role in study design, data collection and analysis, decision to publish, or preparation of the manuscript.

**Competing interests:** The authors have declared that no competing interests exist.

or aid in understanding pathological movement synergies, such as those observed in stroke survivors [6,7]. Additionally, the increasing use of exoskeletons in rehabilitation and industrial settings raises questions about their long-term impact on natural coordination patterns [8–10]. Given its relevance across multiple disciplines, inter-joint coordination remains a central topic in movement analysis.

Inter-joint coordination is inherently complex, involving multiple degrees of freedom and both spatial and temporal relationships. Various metrics have been developed to quantify coordination, but no single approach comprehensively captures all relevant aspects [11]. Broadly, existing methods can be classified into statistical approaches (e.g., Pearson and Spearman correlation coefficients [12,13]), signal analysis techniques (e.g., cross-correlation [14]), and event-based timing metrics (e.g., inter-joint coupling interval [7]). Additionally, kinematic-based methods such as angle-angle plots [15–17] and the covariation plane [18] provide graphical representations of coordination patterns. Two commonly used approaches, Principal Component Analysis (PCA) and Continuous Relative Phase (CRP), stand out for their ability to quantify coordination from different perspectives.

PCA is frequently employed to reduce the dimensionality of joint motion data, allowing for the identification of dominant coordination patterns [19–21]. This technique aims to condense the dataset by identifying a few uncorrelated components that are linear combinations of the original variables (namely joint positions or velocities for current purposes), effectively capturing most of the movement variability (See Fig 1a). By transforming the data into a new coordinate system, PCA enables the description of data variation using fewer dimensions than the initial dataset [22]. By capturing variance in movement strategies, PCA has been used to classify motor synergies and assess differences in movement control across populations [23]. However, comparing PCA results between conditions or individuals remains challenging due to variability in component weights and explained variance distribution. Some approaches have attempted to simplify PCA comparisons by, for example, computing the distance between two reference frames defined by two PCA [24], but they lack explainability from a physiological perspective. Additionally, standard PCA does not account for temporal relationships between joints, limiting its applicability in dynamic coordination analysis [25].

CRP, on the other hand, provides a phase-based representation of coordination by combining position and velocity data into a single measure [26–30] (See Fig 1b). This approach enables the analysis of synchronization and lead-lag relationships between joints over time. Despite its advantages, CRP presents challenges in terms of interpretation and comparison across multiple degrees of freedom, as it typically requires pairwise joint analysis and lacks standardized quantitative comparison tools [31].

To address these limitations, this study introduces two novel indices derived from PCA and CRP to enhance the analysis of inter-joint coordination. The first metric, Joint Contribution Variation based on PCA (JcvPCA), quantifies differences in joint contributions to movement, providing insight into coordination strategy variations. The second metric, Joint Synchronization Variation based on CRP (JsvCRP), emphasizes variations in the temporal synchronization of joint trajectories. By integrating these two complementary approaches, the proposed indices aim to facilitate a more comprehensive assessment of inter-joint coordination.

The remainder of this paper presents the development and validation of these indices using simulated datasets, followed by their application to experimental data collected from a reaching task performed with an exoskeleton motion capture system.

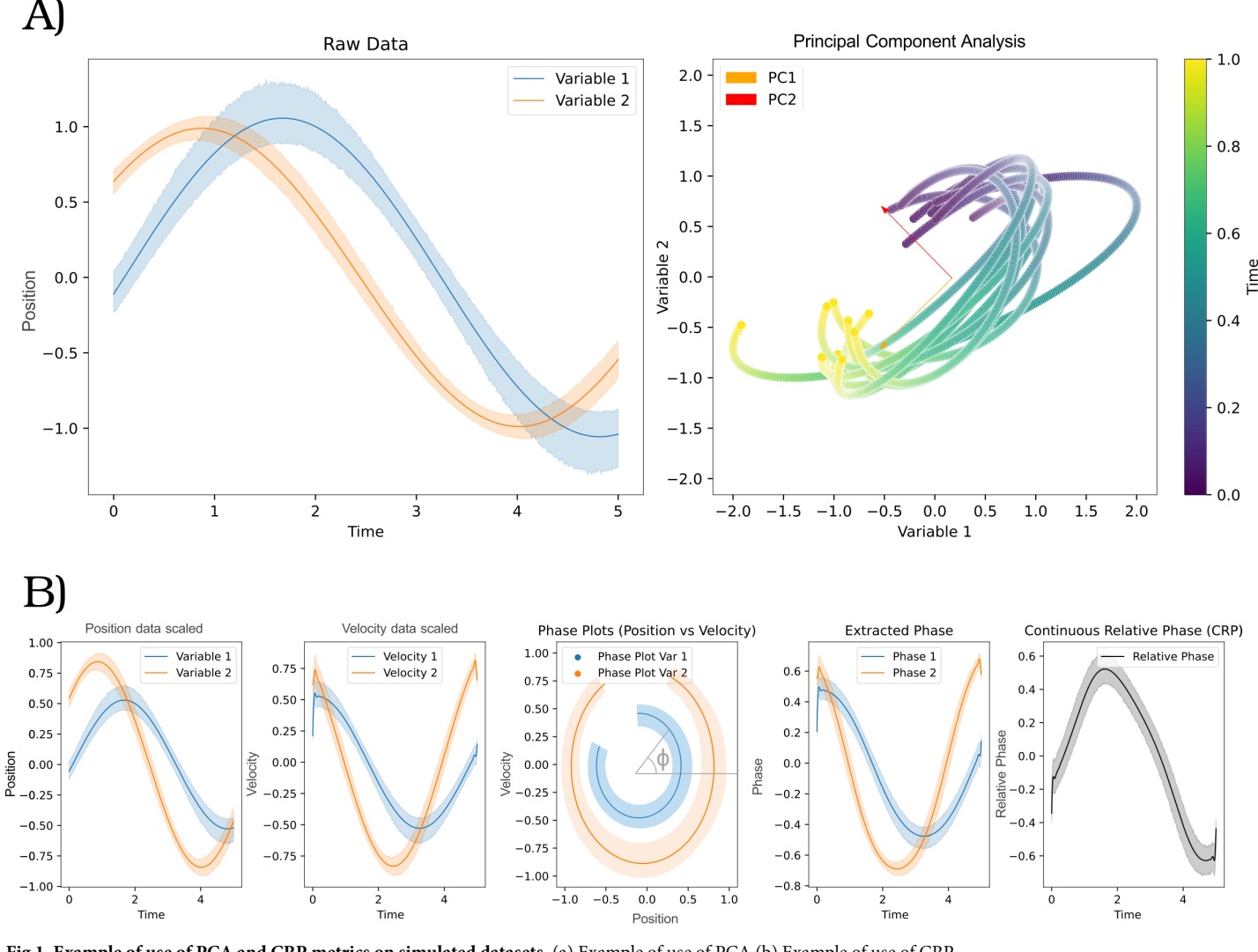

**Fig 1. Example of use of PCA and CRP metrics on simulated datasets.** (a) Example of use of PCA (b) Example of use of CRP.

## 2 Method

### 2.1 Mathematical framework for joint contribution analysis using PCA reprojection (JcvPCA)

The JcvPCA metric enables the comparison of two large datasets containing numerous joint trajectories. It not only identifies the differing joint contributions but also quantifies the extent of those differences. By employing this approach, a more comprehensive understanding of the disparities in joint participation to the movement between the datasets can be obtained.

The following paragraphs outline the four main steps for computing JcvPCA, which is used to compare two datasets consisting of trajectories from $n$ joints while considering $m$ principal components (PCs). We suggest choosing $m$ as $m = p + 1$ where $p$ is the minimum number of degrees of freedom required to perform the task. This process yields a result with dimensions $n \times m$. If needed, all used notations are summarized in Supporting informations S1 File.

**Run PCA on the first dataset.** Datasets A and B are composed of respectively $k$ and $l$ repetitions of the task. One repetition of the task is composed of the evolution in time of the $n$ joint trajectories. Variables $\theta_{A,1}, ...\theta_{A,k}$ are part of dataset A, and variables $\theta_{B,1}, ...\theta_{B,l}$ are part of dataset B. Each $\theta$ contains the $n$ joint trajectories for one movement repetition. Datasets A and B should contain the same number of joints $n$ but do not necessarily contain the same number of repetitions of the task $k$ and $l$. Also, since PCA do not consider the timing of the movement, each repetition of the task can have a different duration. The initial phase involves performing PCA on the first data set. The first dataset will serve as a reference dataset and should be chosen carefully since the results will depend on this dataset. The change in coordination strategy will be determined with respect to the coordination strategy objectified from the initial dataset. The result will be changed if the reference dataset is different since this would infer an alternate initial configuration. A comparison of A with B would yields a different result than the comparison of dataset B with A. The reference dataset must be clearly defined. In our case, dataset A will be the reference dataset.

In the context of inter-joint coordination, before computing PCA, the data are centered to zero (i.e. subtracting the mean of the dataset in order for the new mean of the dataset to be 0) but not normalized to preserve the information from joints that contribute significantly to the task and avoid amplifying noise. Centering data also has the effect of removing the small offsets in the starting position between different datasets. When PCA is performed, each PC obtained represents a linear combination of the joint trajectories. The PC captures the directions in the joint space that account for the most variance in the data. For $u = 1, ..., m$ the corresponding PC is:

$$PC_{A,u} = \sum_{i=0}^{n} a_{u,i}\theta_{A,i} \tag{1}$$

with $a_{j,i}$ is the i-th coefficient of the u-th eigenvector. This first step creates a new frame linked to the dataset A such as : $R^A = \{PC_{A,1}, ..., PC_{A,m}\}$

**Project second dataset in the first PCA space.** The second step consists of projecting the data from dataset B into the $R^A$ frame. This transformation ensures that the data from Dataset B are aligned with the same coordinate system as Dataset A, facilitating further comparison between the two datasets. Thus, any differences or similarities in the joint trajectories between the two datasets can be more easily analyzed. Projecting dataset B in $R^A$ is done for each joint of dataset B such as $i = 1, ..., n$, $\theta_{B,i}^A = R^A \theta_{B,i}$

**Re-compute a PCA on the projected data.** The third step consists of computing a PCA on the projected data $\theta_{B,j}^A$. The PCA returns the PCs with $u = 1, ..., m$ :

$$PC_{B,u} = \sum_{i=0}^{n} b_{u,i}\theta_{B,i}^A \tag{2}$$

where $b_{u,i}$ are the i-th coefficient of the u-th eigenvector. By substituting $\theta_{B,i}^A$ by $R^A \theta_{B,j}$ it becomes possible to express $PC_{B,u}$ in terms of $\theta_{A,j}$ (since $R^A$ is a function of $\theta_{A,j}$) enabling a direct comparison between the expression of $PC_{B,j}$ and $PC_{A,j}$, both expressed in terms of $\theta_{A,j}$. This result is an intermediate result of JcvPCA computation and could be used directly to compare datasets. This intermediate result is called Joint Reprojection Weight (JRW).

**Subtract the weight of joints in each PC.** To make the comparison easier, absolute values of $PC_{A,j}$ can be subtracted from $PC_{B,j}$ to highlight the differences between the two datasets.

JcvPCA result for the $i$-th joint in the $j$-th PC can be expressed as :

$$JcvPCA_{u,i} = |a_{u,i}| - |b_{u,i}^{A}| \tag{3}$$

with $a_{j,i}$ being the weight of the $i$-th joint in the $j$-th PC of dataset A and $b_{j,i}^{A}$ being the weight of the $i$-th joint in the $j$-th PC of dataset B that have been first reprojected in the PCs space of the dataset A.

A positive result indicates that the joint was more used in dataset B than in dataset A while a negative result indicates that the joint was less used in dataset B than in dataset A. So the overall results vary between -1 and 1, a negative result indicating a decrease in the use of the joint and a positive result indicating an increase in the joint contribution to the movements. The specific phenomenon to be characterized will determine whether to concentrate on the first or last PC. If the objective is to examine among the $n$ joints of the user, the ones functionally used to execute the task, the first $p$ PCs have to be used. On the other hand, if the aim is to draw conclusions regarding the use of redundant joints (i.e. within the null space), the last $(m-p)$ PCs will be analyzed.

**Optional: Report results to the explained variance.** To compare the overall results and draw conclusions about the change in coordination strategy, the results obtained at the end of the previous step can be reported to the explained variance of each PC. This can be achieved by multiplying the result obtained for each PC by its corresponding explained variance, such as :

$$res_j = \sigma_j^2 \times (PC_{B,u} - PC_{A,u}) \tag{4}$$

with $(\sigma_1^2, ..., \sigma_m^2)$ being the explained variance of each PC.

Finally, the change of weight of each joint in each PC can be compared and associated to the amount of change in the movement.

## 2.2 Mathematical framework for spatio-temporal joint synchronization using CRP (JsvCRP)

Continuous Relative Phase (CRP) "*is a measure, which describes the phase space relation between two segments as it evolves throughout the movement*" [27]. Unlike other metrics, the CRP takes into account both position and velocity of the segments under analysis. One limitation of the CRP is the comparison of multiple pairs of temporal signals. This new metric is named Joint synchronization variation based on CRP (JsvCRP) and facilitates easier comparison of multiple temporal signals of the initial CRP

**CRP computation.** To compute the CRP, joint position and velocity profiles must be normalized and centered to zero. However, it is important to note that range normalization (see Eqs 5 and 6) can also amplify noise if the range of the noise is larger than the range of the actual movement. Therefore, while range normalization is compulsory to extract a meaningful phase angle, careful consideration should be given to the potential impact of noise amplification. For example, the ratio between the noise of the signal and the actual range of motion of the joint could be computed. If this ratio for one joint is higher than 1 it should be considered that the CRP won't provide meaningful informations when considering this joint. Another possibility is that if this ratio is too high, and that the movement of this joint is residual, instead of normalizing the signal, one solution is to set it as a constant signal that equal to 0 during all the movement. This represents the hypothesis that the joint does not synchronize with the other joints nor contribute to the movement. However, it might not exactly reflect the physiological reality.

$$\theta_{i,norm}(t) = 2 \times \frac{\theta_i(t) - \theta_{i,min}(t)}{\theta_{i,max}(t) - \theta_{i,min}(t)} - 1 \tag{5}$$

$$\dot{\theta}_{i,norm}(t) = 2 \times \frac{\dot{\theta}_i(t) - \dot{\theta}_{i,min}(t)}{\dot{\theta}_{i,max}(t) - \dot{\theta}_{i,min}(t)} - 1 \tag{6}$$

If the goal is to extract a global behavior for a whole set of movements, data can also be normalized in time in order that each CRP then evolves between 0 and 100% of total movement duration, facilitating the comparison of CRP datasets. Different methods can be used for time normalization. If the movement times are similar, basic time normalization by dividing each timestamp by the last timestamp can be sufficient. If the movement times within a dataset are remarkably different or if the relative amount of time for the different parts of the movement are too different, other time normalization methods should be used, such as dynamic time wrapping (DTW) [32], which does not normalize data linearly but tries to align data such that the same number of timestamps in the original data might correspond to different durations in the wrapped data, if that makes the alignment cost smaller. Thus DTW aligns time-series data with varying temporal distortions Another method to align data together is called "registration" [33] that aims to find a transformation (translation, rotation, scaling, etc.) that aligns two datasets spatially or temporally. This method is more general and handles various types of transformations.

To compute the phase angle for each time step, the position, and velocity of each joint are plotted together, creating a phase portrait plot for each joint. For each time step, the position value is represented on the horizontal axis, and the corresponding velocity value is represented on the vertical axis. The phase angle is then defined as the angle between the horizontal axis and the velocity-position point on the plot and can be extracted using the tangent (Eq. 7). This angle provides information about the phase relationship between the position and velocity of the joint at each time step.

$$\phi_i(\theta_i, \dot{\theta}_i) = tan^{-1}\left(\frac{\dot{\theta}_{i,norm}(t)}{\theta_{i,norm}(t)}\right) \tag{7}$$

Finally, phase angle signals are subtracted two by two to extract the CRP between joints.

$$CRP(\theta_i, \theta_j) = \phi_j(\theta_j, \dot{\theta}_j) - \phi_i(\theta_i, \dot{\theta}_i) \tag{8}$$

A constant CRP means that the relation between the 2 joints is constant. A positive CRP indicates that the second joint takes the lead, while a negative CRP indicates that rotation of the second joint would follow those of the first. If needed, all used notations are summarized in Supporting informations S1 File

**JsvCRP computation.** To make the CRP curve comparison easier, a metric that proves to be robust in determining the dissimilarity between CRP curves is by calculating the area between the two mean CRP curves. This metric is named JsvCRP and characterizes temporal discrepancies between the 2 joint's phase angles, and therefore, synchronization. There are $C_n^2$ JsvCRP results for $n$ joints. The JsvCRP can be computed as :

$$JsvCRP_{A,B} = \int_0^{t_{mvmt}} |(CRP_B(\theta_i, \theta_j) - CRP_A(\theta_i, \theta_j))| dt$$

By computing the area between these curves, we can quantify the extent of their differences. A larger area indicates a greater dissimilarity between the CRP curves, signifying more distinct coordination patterns. The area between the two curves is also an easily visualizable indicator, making it simple to extract which parts of the movements differ the most. This approach provides an overview of the differences between the CRP curves, facilitating their comparative analysis.

## 2.3 Data collection for validation of joint coordination metrics

To validate the previously described metrics, two datasets were generated: a simulated dataset, which illustrates the functionality of the metrics, and an experimental dataset which evaluates the metrics during a forward reaching task with healthy adult participants.

**2.3.1 Generation of simulated dataset.** The metrics described above are primarily tested to compare 2 different simulated datasets, named A and B, composed of 2 joints each ($\theta_1$ and $\theta_2$). Datasets A and B are composed of 2 sine waves. $\theta_1$ is the same for both datasets. $\theta_2$ has a phase shift of respectively $1\ rad$ and $\frac{\pi}{2}\ rad$ in dataset A and B compared to $\theta_1$. The amplitude of $\theta_2$ is doubled compared to $\theta_1$ in both datasets (See Fig 3A).

**2.3.2 Acquisition of the experimental dataset.** An experimental dataset was also collected to evaluate the proposed metric upon adult participants performing distinct movement patterns. For this validation, one adult participant of 36 years old and no known neurological or othopedic conditions was recruited. In the experimental protocol, each participant performed reaching tasks using various coordination strategies between the shoulder and elbow. The JcvPCA and JsvCRP methods were then applied to characterize the spatiotemporal features of these different movement patterns.

**Experimental material.** Data collection was carried out using a 4-DoF exoskeleton, Able [34] for controlled motion-capture purposes. The exoskeleton was set in transparent mode for elbow and shoulder movement along the sagittal plane, thus permitting unrestricted flexion/extension of these joints. Rotations in the frontal and transverse planes were blocked via rigid control in order to limit movement along these axes (e.g. shoulder abduction/adduction or internal/external rotation). In this manner, the dataset was reduced to 2 DoF, with the participant performing reaching tasks using exclusively flexion/extension through the shoulder ($\theta_1$) and elbow ($\theta_2$). The wrist of the participant is blocked using a preformed orthosis, restricting the wrist's movements. All movements were recorded using the 1kHz joint position encoders integrated into the robotic exoskeleton.

A screen was placed 2m in front of the participant in order to project 3 distinct targets placed at 3 different heights (Fig 2) The height of the participant's hand was determined using the direct kinematic model of the robot and was visually represented on the screen. In this case, the end point projected on the screen, corresponds to palm height (excluding the fingers).

The task itself involved 1 DoF, where the participant was required to reach the given height of the specified target. Movement of the hand was constrained to move in the vertical plane (2 DoF task) aligned with the participant's shoulder. Each movement began from the same position, with the participant's hand placed at the level of his thigh, shoulder aligned with the body, and elbow in a comfortably flexed position of approximately 140 degrees. To reach the target, the participant freely moved the position of his hand along the vertical plane, to the desired height indicated by the target.

**Experimental protocol.** The participant was asked to reach each of the 3 targets 5 times, using different coordination strategies. The dataset of this experiment can be found in

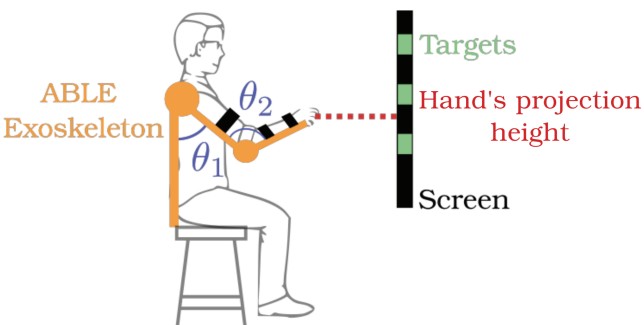

**Fig 2. Experimental set-up.** The participant is wearing the exoskeleton (in yellow) and can use shoulder flexion ($\Theta_1$) and elbow flexion ($\Theta_2$) to reach targets (in green) on the screen in front of them.

S4 Dataset. Participants were asked to perform the task using 4 different coordination strategies in order to modulate both temporal and spatial aspects of inter-joint coordination:

- *Physiological*, in which a participant reached targets with no specific constraints. This was considered the baseline coordination strategy.
- *Temporal desynchronization* in which the participant was asked to move their joints sequentially, first using shoulder flexion and then elbow extension. This condition was used to verify the utility of the novel metrics in characterizing differences in temporal coordination.
- *Single joint* consisted in reaching the target only using one joint, that being the shoulder ($\theta_1$). This condition was used to test a change in both temporal and spatial coordination of joints.
- *Overuse of one joint* consisted in using one joint excessively. In this case, the shoulder ($\theta_1$) was performing the same movement as the *Physiological* condition, while the elbow ($\theta_2$) was first performing flexion and then extension to reach the target. This last conditions was used to test the ability of the metrics to characterize changes in spatial coordination.

The study (number : CER-2023-DUBOIS-Coordination-mouvements) was approved by the local ethics committee Comité d'Ethique de la Recherche de Sorbonne Université, and each participant provided written informed consent prior to his participation in this study. The recruitment for this study and recording of data was done on the 24th March 2023.

## 2.4 Data processing and analysis using joint coordination metrics

For the simulated dataset, the dataset A was used as the reference frame to which dataset B will be compared.

For the experimental dataset, the *Physiological* dataset was used as the reference dataset. But now working with the experimental dataset involves analyzing multiple repetitions of the same movement, which inherently introduces variability into the data. Since the goal of these metrics is to compare two conditions, it is essential to establish a threshold that allows us to determine when the compared dataset is different from the *Physiological* one. Defining this threshold provides a baseline against which the results obtained in subsequent comparisons can be evaluated. To do so, the *Physiological* dataset is shuffled and randomly split in two. JcvPCA and JsvCRP are computed between the 2 subdatasets. These 2 steps of splitting randomly and computing the metrics are performed several time. The obtained result

corresponds to the natural variability of the metrics within a same condition. The *Physiological* dataset was split 15 times into 2 different parts and both JcvPCA and JsvCRP have been computed on the 2 subdatasets, allowing the definition of thresholds for natural variability.

## 3 Results

### 3.1 Validation using the simulated dataset

**JcvPCA Results.** JcvPCA was applied on a the simulated datasets (see Fig 3A). In our case, with only 2 variables, a simple 2D plot representing the evolution of the variables together can be displayed (see Fig 3B). The computation of a PCA on dataset A is performed to extract the weighted coefficients of each variable participating in each PC (equation at the bottom of Fig 3C). The computation of a PCA on dataset B projected in the first PCA reference frame gives the weight of each variable, depending on $PCA_A$. By replacing $PC1_A$ and $PC2_A$ by their expression obtained on Fig 3C, it becomes possible to express $PC1_B$ and $PC2_B$ in terms of $\theta_1$ and $\theta_2$. Fig 3E presents the absolute values of the coefficients of $\theta_1$ and $\theta_2$ for $PC1$ and $PC2$ for both datasets; this is the joint reprojection weight (JRW). The final result of JcvPCA is the subtraction of both PCs' results and is presented in Fig 3F. In this example, we analyze

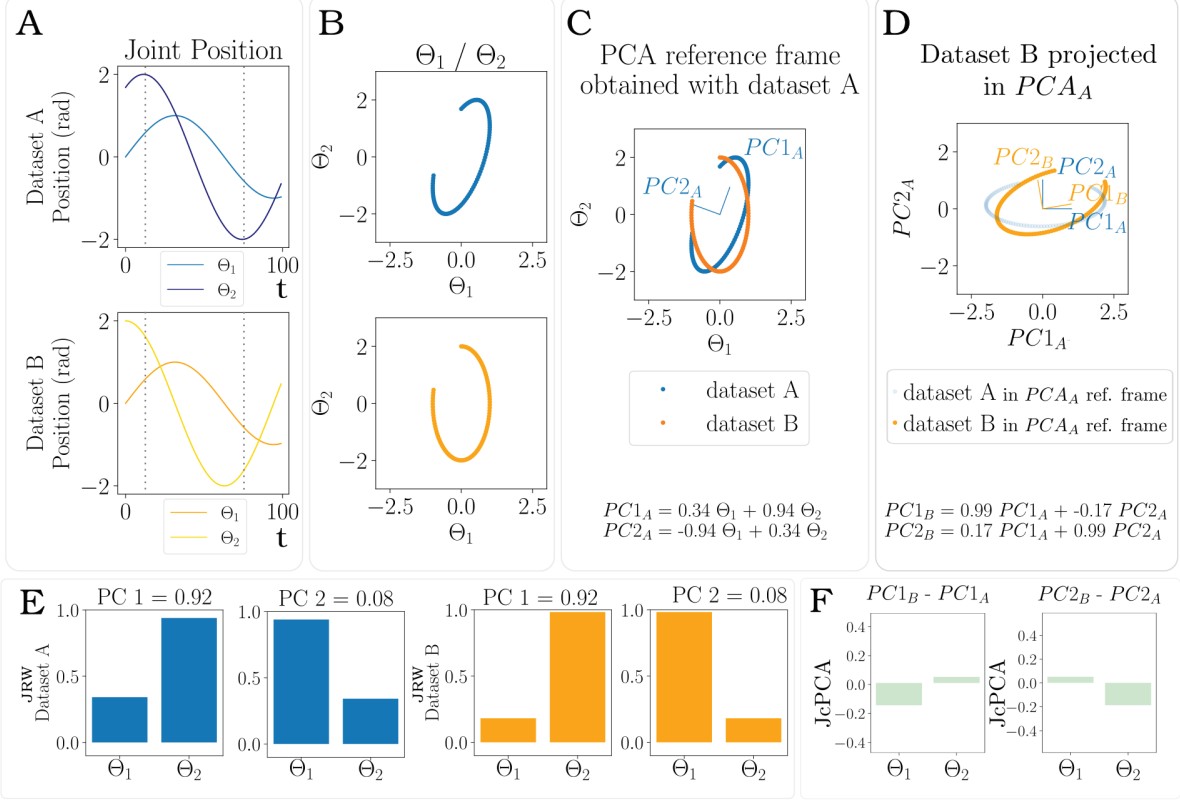

**Fig 3. JcvPCA on simulated data.** (A) datasets pertaining to kinematic time series for 2 joints. (B) Representation of joint positions using angle-angle plots. (C) PCA is computed on dataset A. (D) $PCA_A$ becomes the new reference frame and data of the second dataset are projected in this new reference frame. Another PCA is conducted on the projected data of dataset B. (E) Using equations at the bottom of C and D, the PCs of the second PCA can be expressed in terms of joint position. The coefficient before each joint can be extracted for each PC, this is the Joint Reprojection Weight (JRW). Each PC accounts for a percentage of the total variance of the dataset, but now the PCs of the 2 datasets account for the same percentage. (F) the results for the second dataset is subtracted from the reference dataset.

only PC1 to draw conclusions on the joints that are used to perform the task. As can be seen, $\theta_1$ contributes less to task performance in dataset B compared to dataset A, its contribution decreases by 18%. Conversely, the contribution of $\theta_2$ is slightly greater and is increased by 7%. These results alone are not necessarily telling. To know if this amount of change in the coordination strategy is significant, a baseline measuring the natural variability of movement in the same experimental condition should be conducted (per Sect. 2.3.2).

Were the two datasets exactly the same, the 2 PCA would equally have yielded the same results, hence subtraction of the PCs weights would have lead to a null result for JcvPCA, meaning no measurable change in joint coordination. In contrast, if the 2 variables of the datasets were inverted in dataset B, the PCA reference frame would have been shifted by 90° and the expression of $PC1_B$ would have been depending only on $PC2_A$ and the other way round, showing a complete change of strategy.

In conclusion, the result of this metric emphasizes differences between joint contributions for a given motor task. This may effectively highlight over or under-used joint axes, potentially indicative of altered neurological or musculoskeletal function.

**JsvCRP Results.** JsvCRP was tested on the same simulated datasets. Fig 4A presents the normalized joint position and Fig 4B presents the corresponding normalized velocities. Fig 4C displays the plot of the position depending on the velocity, from which the phase angle will be extracted as the angle between the horizontal axis and the position/velocity point. Therefore, one phase angle signal is computed as shown in Fig 4D and both signals can be subtracted resulting in the dotted line Fig 4D. Finally, the 2 CRP signals of the 2 different datasets can be compared and the area between the 2 curves can be computed (Fig 4E) and used as a metric to quantify the change of coordination between the 2 conditions. The area between the two curves measures 3.06 rad.s. However, this result is not interpretable in isolation. Its significance relies on contextual factors such as the nature of the task. Moreover, the interpretation is contingent upon the natural variability inherent to this particular task, as detailed in the Sect. 2.3.2.

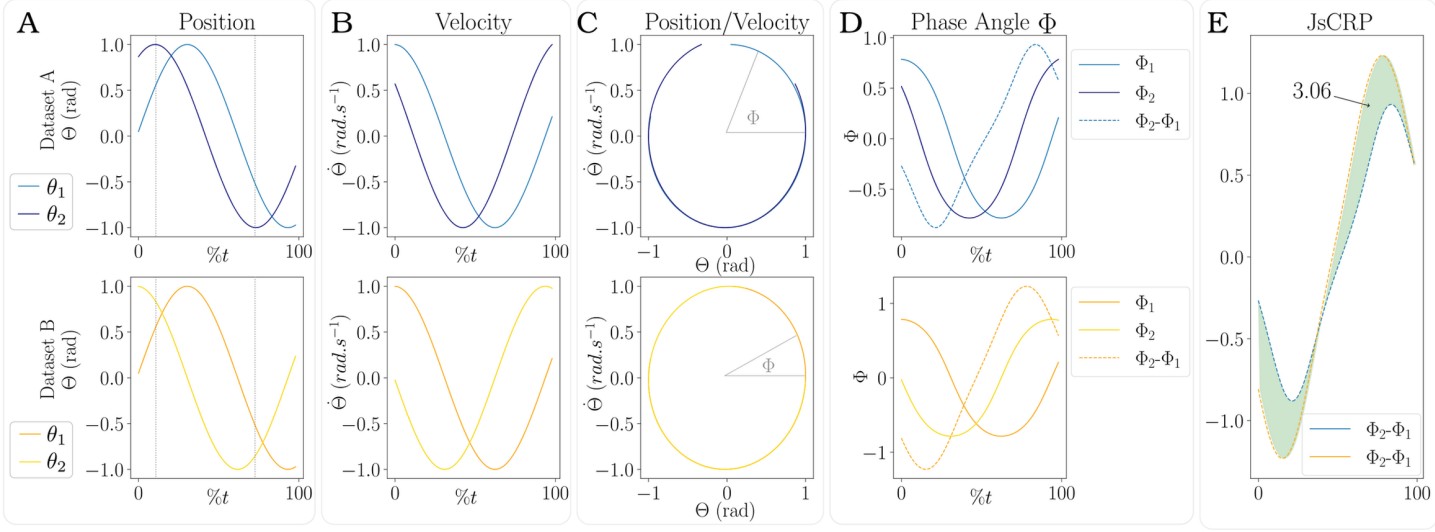

**Fig 4. JsvCRP on simulated datasets.** (A) 2 datasets composed of a time series of 2 joint amplitudes. (B) joint velocities are computed and normalized to their range. (C) joint velocity is plotted with respect to position. (D) the angle between the velocity-position point and the horizontal (zero-velocity) axis is extracted for each timestamp. (E) The JsvCRP is calculated as the difference between the phase angles of two joints and the area between the 2 curves is computed.

In conclusion, the JsvCRP, defined as the area between 2 CRP curves, provides a valuable indication of the extent of the changes in coordination strategy. A larger area between the curves indicates a more substantial difference in the joint coordination patterns.

An example python code is available for download to test both metrics with this simulated dataset in the Supplementary S2 File.

## 3.2 Validation using the experimental dataset

Fig 5 presents the 4 different datasets recorded using the different inter-joint coordination strategies presented in Sect. 2.3.2. The large variability at the end of the movements is due both to the height of the different targets and to the natural variability of the subject. An animation replaying the recorded strategies can be downloaded in the Supporting Information section as S3 Video.

Previously described metrics are computed on each distinct coordination strategy employed by the participant during the reaching tasks with respect to values obtained for the reference, i.e. the *Physiological* coordination strategy. The metrics can be numerically compared together since they come from the same experimental protocol and the reference dataset (dataset A, i.e. the *Physiological* dataset) is the same for all comparisons.

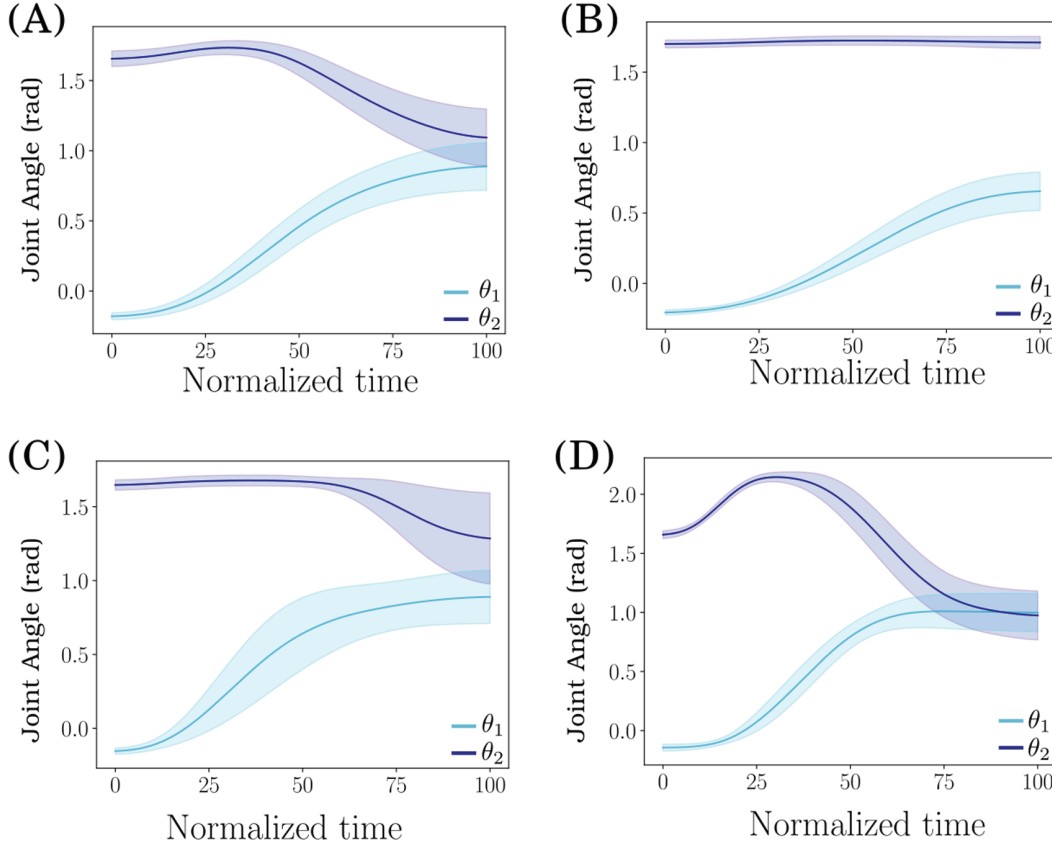

**Fig 5. Four coordination strategies using shoulder flexion and elbow extension.** (A) Physiological Coordination Strategy. (B) Desynchronization of the 2 joints. (C) Use of the shoulder only. (D) Overuse of the elbow. The mean trajectory as well as the standard deviation are presented.

**Natural Variability.** As presented in Sect. 2.3.2, the threshold of both metrics, due to natural variability is computed over the *Physiological* dataset. The average value of JcvPCA computed solely over the *Physiological* dataset is $-0.004 \pm 0.03$ for the shoulder flexion and $0.007 \pm 0.05$ for the elbow flexion. The average value of JsvCRP computed solely over the *Physiological* dataset for the shoulder and elbow synchronization is $622.3 \pm 418.6$ deg.s.

These findings indicate that, within the scope of this experiment, when comparing two datasets, if the resultant values fall within these intervals it is inconclusive to infer that the datasets encompass disparate coordination strategies. However, if the obtained result is outside this interval, there may have been a change in coordination strategy.

**JcvPCA Results over experimental datasets.** JcvPCA is computed on each dataset, with the *Physiological* dataset serving as the reference (in blue on Fig 6). The other coordination strategies were then reprojected into the *Physiological* PCA space. The left panel of Fig 6 illustrates JRW and results of the JcvPCA are presented in the right panel.

For the overuse of the elbow strategy, results illustrated across the fourth row of Fig 6, indicate that the contribution of the elbow (joint 2) to PC1 appears proportionally greater than the same joint in the physiological movement (per JRW representation). This corresponds with a positive value for the elbow in the JcvPCA result indicating an increase of around 18% in the contribution of the elbow to the movement. In contrast, there is a decrease in the use of the shoulder joint by 13%.

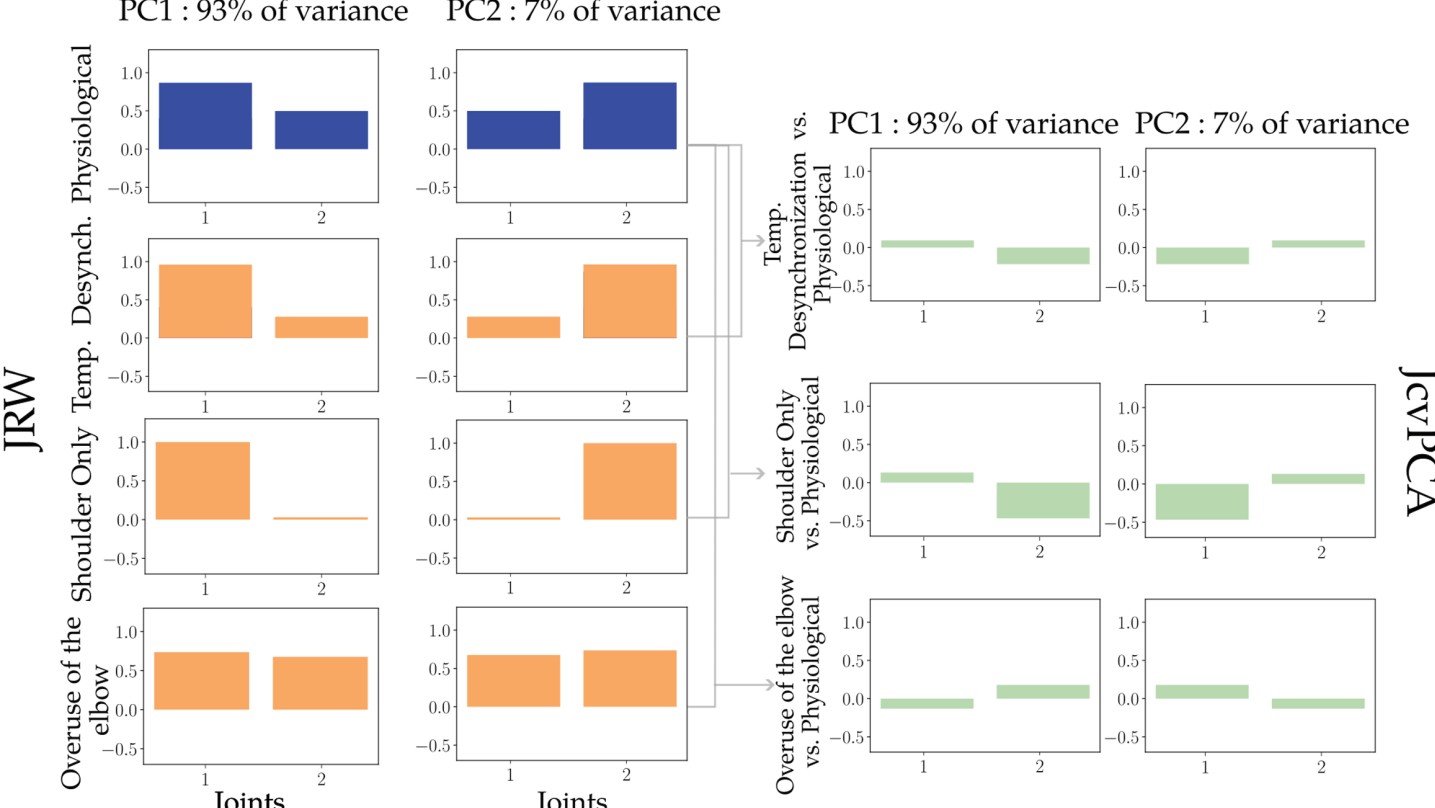

**Fig 6. JcvPCA results on experimental datasets.** Each coordination strategy data has been reprojected into the Physiological PCA reference frame and the Joint Reprojection Weight (JRW) are reported on the left. JRW of the of the physiological coordination strategy are indicated in blue, while divergent coordination strategies are indicated by orange bars. In the right panel, JcvPCA for each coordination strategy, with respect to the Physiological coordination strategy, are represented in green.

For the second coordination strategy, the *Shoulder Only* coordination, results across the second last row indicate that shoulder contribution (joint 1) to PC1 is greater than for the *Physiological* condition. At the same time, a marked reduction in the contribution of the elbow is indicated in the JcvPCA with a value of -0.4 for the change of contribution of joint 2. Conversely, the opposite trend is observed in the second PC. This result can be interpreted as a decrease of 47% in the contribution of the elbow and in contrast an increase of 13% in the use of the shoulder in the first PC, that contributes to the task execution.

Finally, for the *Temporal Desynchronization* condition, in the first line of the Fig 6, in the first PC, elbow rotation ($\theta_2$) is used less than in the *Physiological* condition. This might be an artifact of the protocol, as the contribution of the shoulder rotation may have increased given that the subject was instructed to use this joint exclusively through the initial stages of the movement. Beyond this observation, values of the JRW remain comparable to the physiological condition. As indicated above, this is to be expected, as PCA has limited capacity to enhance temporal differences in movement strategies. In this case, the variation of the shoulder contribution is of 9%, thus just above the significance threshold used as a baseline. The contribution of the elbow is increased by 22%, mainly due to the fact that human subjects are not good at only desynchronizing joints while keeping the same contribution.

In summary, these results illustrate how the JRW and JcvPCA metric might effectively capture differences in the contributions of different joint axes to a given movement. As such, the JcvPCA might serve in elucidating distinct coordination strategies or highlight changes over time. However, JcvPCA is not a proper tool to evaluate changes in joint synchronization.

**JsvCRP Results over experimental datasets.** CRP was computed over all trials of all targets, and the mean CRP was extracted. Fig 7 displays the mean CRP between joints $\theta_1$ and $\theta_2$ for the different coordination strategies. The blue curve indicates the *Physiological* coordination strategy. The area between the curves, highlighted in green, was computed and used to quantify differences between the two CRP curves.

For the *Temporal Desynchronization* strategy, JcvCRP equals 2411 units of deg.s. This value is well above the natural variability threshold, showing a difference of synchronization in joints. The CRP curve exhibits similarity at the beginning and end of the movement. This can be explained by the fact that, when reaching a target, naturally, the shoulder joint initiates the movement, and towards the end, the elbow joint is used for fine adjustments and corrections of the end-effector position. However, between 40% and 80% of the movement, the CRP

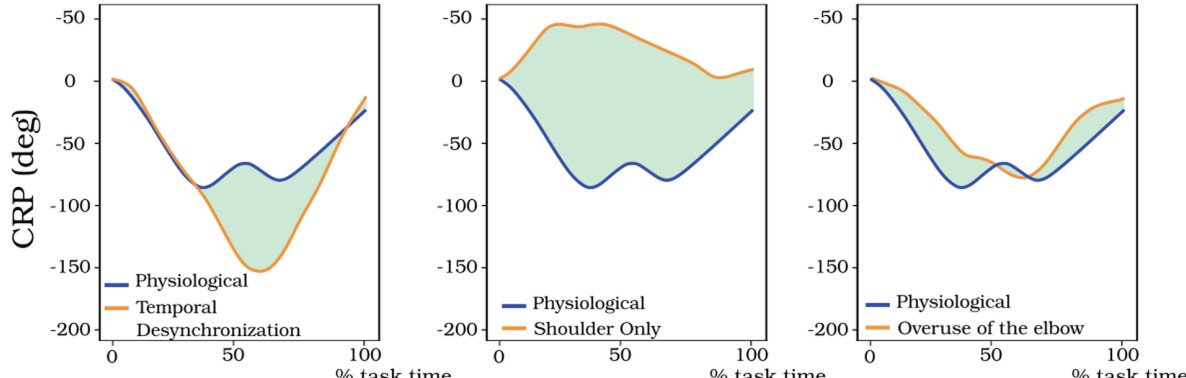

**Fig 7. CRP results between joints 1 and 2 for the 3 different coordination strategies, compared to the physiological CRP (in blue).** In green is the area between the 2 curves.

curves diverge significantly. The CRP of the *Temporal Desynchronization* strategy continues to decrease during this phase and then increases when the shoulder joint ceases its movement and the elbow joint takes over. In the *Physiological* coordination strategy, both joints are moving together for around 50% of the movement, creating this wave shape.

Regarding the *Shoulder Only* strategy, JcvCRP equals 8272 deg.s. Once again, this value is far higher than the significant threshold, indicating an important change in coordination strategies. The CRP curves for the two joints are entirely different because the elbow joint remains stationary throughout the movement. This absence of movement in the elbow joint leads to a distinct CRP pattern compared to the other strategies.

For the *Overuse of the Elbow* coordination strategy, the CRP is similar to the *Physiological* coordination strategy when speaking about the position/velocity relation between the joints. The area between the two curves is 1787 units, indicating a change of joint synchronization since this value is above the natural variability threshold, however, the difference of synchronization is way smaller than for the two other coordination strategies. The main differences are at the beginning and at the end of the movement, where the elbow is used more than in the *Physiological* condition.

In conclusion, the JsvCRP technique captures variations among various coordination strategies and proposes findings that can be understood from a physiological standpoint. The results from the JsvCRP metric provide specific insight into temporal changes in joint coordination enabling direct comparison's between coordination strategies.

## 3.3 Combined application of JsvCRP and JcvPCA metrics

We introduced metrics to compare the variation of inter-joint coordination between two datasets, from 2 points of view: joint contribution (with JcvPCA) and temporal coordination (with JsvCRP). The metrics described in this paper were applied to an experimental dataset consisting of four different coordination strategies. The JcvPCA provided specific insights into the amplitude of the different axes involved, while the JsvCRP analysis provided information about the temporal aspects of the position/velocity relationships between those joint axes throughout the movement task. These metrics have been developed concurrently with the intent of capturing changes in each respective domain (i.e. joint amplitude, movement timing). As such, analyzing them together offers a more comprehensive perspective on movement variations, enabling a nuanced assessment of inter-joint coordination.

In the case of the *Overuse of the Elbow* strategy, the JcvPCA results showed an increase in the participation of the elbow joint in task execution. The temporal differences, measured with JsvCRP were relatively minor, and primarily observed at the beginning and end of the movement, where the elbow played a more important role, compared to the *Physiological* coordination strategy.

For the *Shoulder Only* coordination strategy, there was a significant shift in joint participation, as shown by the JcvPCA, with the majority of the movement being achieved through shoulder flexion. Additionally, since one joint remained static, the temporal relationship between the joints was completely altered, as captured by a comparably large value for the JcvCRP unit measure.

Finally, the *Temporal Desynchronization* strategy resulted in decreased use of the elbow joint, but more importantly, it revealed different coordination patterns between the shoulder and elbow. The most significant differences were observed in the middle portion of the task, between 40% and 80% of the movement duration. This can be explained by the fact that when performing reaching tasks people tends to use their joints in a proximal-to-distal order [35] [36], [37], leading firstly with the shoulder, then adjusting with the elbow during

task completion, as observed in the *Physiological* condition. Thus, the beginning and the end of the movements of the *Physiological* and *Temporal Desynchronization* conditions are similar.

The concurrent use of both metrics is crucial to achieve a comprehensive understanding of all facets of inter-joint coordination. If the study focuses on a single aspect of inter-joint coordination, either the JcvPCA metric, for evaluating joint contribution variation, or the JsvCRP metric, for assessing joint synchronization variation, may suffice. For instance, in an ergonomic evaluation where the objective is examine joint solicitation under different conditions, the JcvPCA metric alone may be adequate. Similarly, in sporting applications, JcvPCA might provide insight into the relative contributions of different joints when training form or technique. Conversely, in gait analysis involving prosthetics, where tuning the device for the patient predominantly requires ensuring the synchronization of lower-limb joints, the JsvCRP metric alone may be sufficient.

## 4 Discussion

The objective of this paper was to present a novel set of metrics for comparing two kinematic datasets for a given movement task. The two metrics we describe extend upon PCA and CRP, methods which have previously been employed for characterizing coordination strategies in healthy and pathological populations. More specifically, the JcvPCA and JsvCRP which we propose, facilitate valid comparisons between kinematics datasets. Each provide specific values indicative of differences in either the amplitude of joint contribution (JsvPCA) or the timing of joint rotations (JsvCRP). These metrics offer physiological insights into the evolution of inter-joint coordination, surpassing the capabilities of other known metrics, as far as our current understanding extends. We anticipate that these measures may provide the basis for a quantitative approach to measuring differences in inter-joint coordination, yielding valuable insights into physiological movement patterns. In the following discussion, we examine specific aspects to be considered when employing JcvPCA and JsvCRP, as well as perspectives for future applications of these metrics.

**4.0.1 Implementation of the novel metrics.** The datasets examined in the present paper, both simulated and experimental, were composed of two degrees of freedom, with rotation along the sagittal axis for the shoulder and elbow joints. This decision was made for illustrative purposes only, and was intended to provide contrast for specific variations in the amplitude and timing of the paired joint rotations. Nevertheless, implementing the JcvPCA and JsvCRP over a greater number of degrees of freedom remains relatively straightforward. In doing so, the principal consideration for the JcvPCA is to define the number of PCs required (depending on the number of DoF of the task and on the phenomenon to be observed), while for JsvCRP, pairwise comparisons of all rotational axes should be included in the analysis. Alternatively, for situations with a considerable number of degrees of freedom, it might be useful to define paired joint axes that would be the most pertinent, depending on the movement task. For example, with a 1 DoF task (reaching a predefined height), using a 7 DoF model of the arm, JcvPCA could be computed with 2 PCs, the first one containing the variation in joint contribution relative to the task, and the last one containing the variation in joint contribution relative to the null space. With the same example, JsvCRP would contain 21 different results (i.e. 21 possible pairs of DoF with a total of 7 DoF). However, if the task only requires attaining a specific height, one might consider that sagittal plane movement, including shoulder flexion and elbow extension, would contribute most to the task, and thus warrant analysis over other potential combinations that would contain less information regarding the task execution.

In addition to the DoF, the number of repetitions which makeup the dataset is another factor which should be considered. In effect, carrying out PCA is contingent upon having an adequate sample of movements upon which this data compression technique might be valid. This is even more true if the baseline is computed based on one of the 2 datasets before comparing the datasets together, since it's necessary to split the first dataset into sub-datasets. Large datasets imply many participants and/or many repetitions (e.g. reaching different targets), and can lead to long experimental procedures. Many studies have tried to determine how much data is needed to compute a PCA [38]. Usually a variable-to-factor ratio between 5 to 20 is recommended, depending a lot on the study. That means that with 2 degrees of freedom, a minimum of $2 \times 5 = 10$ to $2 \times 20 = 40$ data points are recommended. In movement analysis, usually, the recording of one movement far exceeds this number of points. However, it's good to keep in mind that as the number of variables to be considered increases, so too should the number of movement repetitions to be analysed. One method to check if the number of data in the PCA is sufficient is to bootstrap or cross-validate the PCA result by exchanging or deleting a small fraction of the original data. If the result of the PCA on the bootstrap dataset is similar to the first PCA result, that means that the PCA result is stable and that there is enough data.

The procedures used for movement capture and the calculation of joint angles may also have important implications on the results obtained. Experimental data generated for the present paper was generated using data obtained via the joint position encoders integrated into the exoskeleton (i.e. measuring joint angles of the exoskeleton itself). More commonly, kinematic analysis of human movement tends to be based upon recordings obtained using other techniques (e.g. optoelectronic devices, inertial measurement units) which imply different constraints for approximating joint positions. Furthermore, different conventions exist for the extraction of joint angles from kinematic data. For example, values derived using the calibrated anatomical model proposed by the International Society of Biomechanics (ISB) [39] would not necessarily yield the same results as data extracted using another Euler sequence (as was the case for experimental data here using the exoskeleton). When using JcvPCA and JsvCRP, care should be taken to calculate values on datasets extracted using identical procedures to ensure valid comparisons.

**4.0.2 Specific considerations for JsvPCA and JsvCRP.**   As already mentioned in Sect. 2.1, running JcvPCA from dataset A to dataset B, will provide a different result than running JcvPCA from dataset B to dataset A. This is due to the reprojection in the first PCA reference frame part. Evidently, if both datasets remain in their original reference frame, direct comparisons of PCs weight only would not be valid. Choosing carefully a reference dataset, named A here, such as it is a "standard" or "baseline" condition that will be used as a reference for all the datasets comparison is a key point in obtaining interpretable results.It is important to note that JcvPCA is a suitable metric for monitoring the evolution of coordination strategies. Due to the reprojection step, if the compared strategies are entirely different (not just a variation or an evolution), the reprojected data may lose crucial information specific to each strategy, making the differences less apparent in the results. In cases where strategies differ significantly, especially when considering a large number of joints (which was not the case in this study), a direct comparison of PCA without reprojection would be more appropriate, even if the physiological interpretation could be more complex.

One of the key points with using JcvPCA is the number of PCs to be considered in the analysis. In the examples presented above, the movement task comprised only two joint axes. Accordingly, PCA was based upon 2 PCs and thereby captured 100% of the total variance in the dataset. However, with the addition of further joint axes, it may become impractical to

analyze a number of PCs equal to the number of the measured joint axes. In common practice, the number of PCs required to account for 80% of the variance are analyzed. Based upon this perspective, it may have been sufficient to examine the first PC, accounting for 93% of overall variance in the examples described here. Based upon the properties of the specific dataset, it may be necessary to analyze several PCs (e.g. 4 or more) in order to have a sufficient sample for analysis. For example, in a task that requires 3 degrees of freedom, at least 3 PCs might be needed to explain at least 80% of the total variance. If the study is primarily interested in how the null space is used, adding one more PC (so 3 PC for the 3 DoF task plus 1 for the null space) might be helpful to characterize the use of the null space. However, increasing the number of PCs will, of course, increase the number of indicators which must be compared when characterizing changes to the movement strategy. The balance between obtaining a physiological result versus having numerous indicators to monitor must be found for each experiment, depending on the goal of the study. For example, if the physiological explanation of the coordination strategy is less important, maybe reducing the number of PCs may streamline the analysis. On the other hand, if a physiological understanding of the coordination patterns employed is the primary object, more PCs should be considered.

The main limitation of CRP is that it may tend to accentuate noise in kinematic data, especially for joint axes with relatively minor participation in the overall movement task. As the normalization process changes certain dimensions of the dataset, it is important to use both JsvCRP and JcvPCA together to produce a coherent perspective of the data at hand. Moreover, CRP is a metric that can be computed using different methods (different normalization processes, and different phase extraction methods such as Hilbert transform [27]). Accordingly, results for the CRP calculations may slightly vary depending on the computation method used, with potential implications upon one's interpretation of joint synchronization across the movements.

An essential consideration in measuring inter-joint coordination, particularly for these metrics, is the definition of the starting position. It must be meticulously defined and is an integral aspect of the task, as a significant alteration in the starting position is analogous to a change in task conditions. To illustrate this, initiating movement above or below the target yields distinct coordination requirements for the joints. In the former scenario, extension of the elbow is necessary, whereas in the latter, flexion of the elbow is required. These variations in joint use yield divergent results from an inter-joint coordination perspective. However, in experimental settings, minor shifts in the starting position may occur. Nevertheless, these slight shifts do not influence the results significantly. Both metrics address this issue through centering (for JcvCPA) or normalizing (for JsvCRP) the datasets, effectively removing the small offsets due to variations of the starting position.

Another point is that, in this paper, JsvCRP is defined as the area between the 2 curves. With our datasets, this metric is a good balance between keeping interesting information and being explainable. Indeed, JsvCRP keeps as much information as possible from the CRP while reducing the temporal curve to a single result that can still easily be analyzed in a physiological manner. Other indicators could also be used in order to keep more or less temporal information. A first step could be to use the mean CRP [31], another interesting method could be to use cross-correlation, already used to directly analyze joint trajectories [14], to analyze these temporal signals.

Finally, it should be remembered that CRP only gives temporal information regarding a normalized timescale. The JsvCRP metric can be used for similar movement times, but if the movements' durations are markedly different, the interpretations drawn from the CRP curves must be handled carefully. One first solution could be to use dynamic time wrapping [40] to counteract this limitation.

### 4.0.3 Natural variability vs. change in coordination?

The metrics described here are, by design, intended for comparisons between 2 datasets. Of course, human movements are seldom exactly the same. While certain features remain comparable, what is observed is often dubbed "repetition without repetition" where each gesture implies unique neural and motor pattern [41] [42]. The issue thus becomes how one distinguishes when changes in coordination metrics reflect this natural variability, or indeed, if it represents a meaningful change in behaviour or the underlying function in the neuromotor apparatus. In effect, no fixed threshold exists to assist in determining whether the shift in a given metric is indicative of a transition between two coordination strategies.

A potential solution to this problem would be establishing such a threshold based upon the variability observed both within and between subjects from a given dataset. To compute inter-subject variability, metrics can be computed multiple times on different subsets of the baseline condition. To compute intra-subject variability, metrics would then be computed for all subjects within the same condition. From those previous results, the mean and the standard error (standard deviation divided by the square root of the number of subjects) of the metrics can be extracted and used to characterize the natural variability. Finally, when computing the metric on datasets of 2 different conditions, if the result exceeds the interval given by the mean plus or minus the standard error, the difference could be considered significant.

By using the mean with the standard error as a threshold, a shift in movement patterning could be categorized as being either a change of coordination strategies, or more simply, the natural variability of those subjects. This natural variability threshold should be recomputed for each experimental protocol, since the condition of the experiment and the task could influence its value.

### 4.0.4 Perspectives for movement analysis in sport, ergonomics and clinical settings.

The JcvPCA and JsvCRP metrics might prove valuable in a range of applications in human movement analysis. To begin with, these approaches may be used to objectively characterize coordination patterns exhibited by people with varying levels of skill. In such a manner, these metrics could be applied to performance enhancement in sporting gestures. For example, highly skilled tennis players with an effective service type (e.g. flat, slice, kick) might be identified. Using JcvPCA and JsvCRP in comparative movement analysis with less skilled players could then be carried out to better distinguish how specific patterns of movement amplitude and timing contribute to the variables of interest (e.g. velocity, spin, etc.). Using this process, the role of joint contributions in determining ball trajectory may be deduced, and the temporal patterns of joint rotations contributing to overall performance might be identified. These observations could then be used to assist players in refining their service action [43].

Perhaps most importantly, the novel metrics we propose are particularly adapted to evaluating change in coordination patterns. As a result, JcvPCA and JsvCRP can be used to determine effectiveness of specific interventions. Within the field of ergonomics, new devices or work procedures might be examined in terms of their impact upon the user's activity. By evaluating specific tasks (e.g. lifting, tool use) before and after, the metrics presented in this paper may reveal how the integration of those tools and equipment influences joint loading. If the integration of that tool fails to solicit change in task performance, JcvPCA and JsvCRP metrics should indicate the absence of change in inter-joint coordination. Any detected changes might indicate the emergence of a novel coordination pattern induced by the equipment. Such alterations, if persistent over time, could potentially have possible negative consequences on the musculoskeletal system of the individual. Thus, this type of procedure may be imperative for ensuring the safe integration of highly advanced assistive technologies, such as exoskeletons, which have the specific vocation of improving physical capacities in industrial settings. While

exoskeletons may improve certain postural configurations, they may equally trigger unanticipated movement compensation [44] [8]. Such devices are today evaluated mostly using EMG signals or heart rate data [45], [46], adding measurable data regarding these changes to user coordination would assist in adjusting feedback parameters.

Within clinical settings, JcvPCA and JsvCRP could be used for monitoring change over time. In physical rehabilitation, these metrics might represent important outcome measures for people suffering from either musculoskeletal or neurological pathologies. In hemiparesis, for example, one of the characteristic traits is the abnormal coupling of different joint axes. In reaching actions, excessive activation of shoulder abductors and internal rotators diminish the habitually smooth coordination as the person moves their arm forwards. In this type of situation, increased weighting on shoulder flexion and elbow extension in JcvPCA (with corresponding decrease in shoulder abduction and internal rotation) would be a direct indicator of progress in rehabilitation. Using the JsvCRP, motor recovery would be expected to mimic the physiological inter-joint coordination described here (per Sect. 3.2) with a wave function indicative of the a movement initiated with shoulder flexion and adjusted through elbow extension. Another possibility would be the addition of these metrics to existing clinical scales. For instance, evaluation of volitional movement synergies with the Fugl Meyer assessment simply provides a simple grading on a 3-point scale (none, partial, full). The integration of JsvCRP could be used to quantify, in terms of desynchronisation, those movement compensations which occur between the paired joint axes which are evaluated. This would provide much greater sensitivity to subtle but important changes in motor control during recovery.

## 5 Conclusion

The two metrics described in this paper have been specifically designed to provide greater perspective regarding inter-joint coordination. Used in tandem, the JcvPCA and JsvCRP can be used to compare specific differences in joint contributions to a given movement task, as well as the variations in temporal synchrony between the joint axes. In JcvPCA, the first dataset undergoes PCA, and the second dataset is projected into the new reference frame defined by the first PCA. By computing PCA on the reprojection of the second dataset in the reference frame of the first PCA, it becomes possible to compare the evolution of the contribution of each joint in each PC. This extension of PCA provides a direct comparison of joint participation without having to consider the percentage of variability within each components (the primary obstacle when comparing two PCAs with existing methods). The second metric is JsvCRP and uses CRP to assess the temporal evolution of coordination patterns. To quantify the dissimilarities in CRP curves, the area between the mean curves of the two CRP is computed and represented. Importantly, both the JcvPCA and JsvCRP convey variation in coordination strategies as a single value. These metrics might thus be directly used in statistical analysis to identify differences in motor behaviour between cohorts, examine participant responses to a specific experimental condition, or document evolution of in movement patterns during the course of an intervention. Finally, each metric is relatively easy to compute and provides results that can be directly interpreted in terms of change to the physiological movements, providing valuable insights into the coordination strategies employed during different task conditions.

## Acknowledgment

Not applicable

## Supporting information

**S1 File. Mathematical notation.** This file contains the details of all mathematical notations used in this article.
(PDF)

**S2 File. Testing Code for JcvPCA and JsvCRP.** This file contains the code that implements both metrics in python and apply them on a simulated dataset.
(ZIP)

**S3 Video. Coordination strategies animations.** This is a video of an animation of the 4 different coordination strategies (*Physiological*, *Shoulder Only*, *Overuse of the elbow*, *Temporal Desynchronization*) that have been experimentally recorded, replayed with a stick figure.
(ZIP)

**S4 Dataset. Dataset of the 4 conditions experimentally recorded**. This zip file contains one folder for each condition. For each condition, the 3 repetitions of the movements for the 3 different targets' height are presented in individual csv files.
(ZIP)

## Author contributions

**Conceptualization:** Océane Dubois, Nathanaël Jarrassé.

**Data curation:** Océane Dubois.

**Formal analysis:** Océane Dubois.

**Funding acquisition:** Nathanaël Jarrassé.

**Investigation:** Océane Dubois, Nathanaël Jarrassé.

**Methodology:** Océane Dubois, Ross Parry, Nathanaël Jarrassé.

**Project administration:** Nathanaël Jarrassé.

**Software:** Océane Dubois.

**Supervision:** Ross Parry, Agnès Roby-Brami, Nathanaël Jarrassé.

**Validation:** Ross Parry, Agnès Roby-Brami, Nathanaël Jarrassé.

**Visualization:** Océane Dubois.

**Writing – original draft:** Océane Dubois.

**Writing – review & editing:** Océane Dubois, Ross Parry, Agnès Roby-Brami, Nathanaël Jarrassé.

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
