## [Decision Letter · Decision Letter 0]

16 Apr 2024

PONE-D-24-06791JcvPCA and JsvCRP : a set of metrics to evaluate changes in joint coordination strategiesPLOS ONE

Dear Dr. Dubois,

Thank you for submitting your manuscript to PLOS ONE. After careful consideration, we feel that it has merit but does not fully meet PLOS ONE’s publication criteria as it currently stands. Therefore, we invite you to submit a revised version of the manuscript that addresses the points raised during the review process.

Please address all Reviewers' comments, and particularly those referring to reporting advantages and limitations of using this method, rather than others.

We look forward to receiving your revised manuscript.

Kind regards,

Roberto Di Marco, PhD

Academic Editor

PLOS ONE

Journal Requirements:

"ANR EXOMAN (ANR-19-CE33-0009) "

3. Please expand the acronym “ANR” (as indicated in your financial disclosure) so that it states the name of your funders in full.

4. We noted in your submission details that a portion of your manuscript may have been presented or published elsewhere. [Figure 1 and 2 have been submitted in a paper accepted to ICRA 2024, representing a preliminary aspect of this study. The figures are used to give a quick overview of the metrics used without going into details.] Please clarify whether this [conference proceeding or publication] was peer-reviewed and formally published. If this work was previously peer-reviewed and published, in the cover letter please provide the reason that this work does not constitute dual publication and should be included in the current manuscript.

Reviewers' comments:

Reviewer's Responses to Questions

**Comments to the Author**

1. Is the manuscript technically sound, and do the data support the conclusions?

Reviewer #1: Yes

Reviewer #2: Yes

2. Has the statistical analysis been performed appropriately and rigorously? 

Reviewer #1: I Don't Know

Reviewer #2: Yes

3. Have the authors made all data underlying the findings in their manuscript fully available?

Reviewer #1: No

Reviewer #2: Yes

4. Is the manuscript presented in an intelligible fashion and written in standard English?

Reviewer #1: Yes

Reviewer #2: Yes

5. Review Comments to the Author

Reviewer #1: A very interesting approach to converting positioning systems. Congratulations on your ideas, but I have doubts about the form of testing and comparing the results.

The article was written correctly and does not raise any objections, except for some incomprehensible issues that were included in the comments section.

Comments:

1. Please add numerical values, a description of the results and conclusions to the abstract, the abstract does not have the correct form - it needs to be reworded

2. Is examining such a small number of people objective? please justify your answer?

3. Can the presented data be compared numerically? please justify your answer

4. The results do not properly present the numerical values obtained during the study - the presentation of the areas under the graphs does not properly reflect the differences? Were average values used for comparison? why average if the sample is so small?

5. Why were these particular strategies chosen to execute the moves?

6. what does it mean that the curves are different from each other? Do we have any numerical values confirming that there are differences? when are the differences significant?

7. Could you point out the advantages and limitations of the methods used, and indicate when to choose each method?

This type of approach will increase the usefulness and citation of the manuscript

Reviewer #2: In this paper, the authors present two new measures to look at coordination between joints, JcvPCA and JsvCRP, which focus on the amplitude of joint contribution or the timing of joint rotations. While the methods presented are likely to be helpful, a limitation of the paper is that it is difficult to understand what a particular value of a score means, e.g. is 8272 a large value for JsvCRP? Also, it is not clear how to use this technique in situations with large numbers of degrees of freedom (as is the case in more naturalistic movements) where many decision need to be made (e.g. how many PCs to use)

Abstract - fix "relatinshipTs" (T instead of ')

page 2 - delete "the" in the Principal Component Analysis, and "the" continuous relative phase

page 3 - For example, [38] presented a metric - could you give some more details about what this metric is based on?

- [5] have introduced function PCA - they did not introduce it, it has been used in many previous works, e.g. the book of Ramsay and Silverman in 1997

"considers one joint at a time" - this is not necessarily the case, see the PCA examples in this book (and their 2009 book on Functional data analysis with R and MATLAB)

CRP - "if the result is constant, the joint phases evolve in unison" - it is not obvious (to me) why this is the case, it would be helpful to explain why this is the case (or maybe provide a graph as an example)

page 4- "there is at least as many joints" → "there are"

page 5 - "k and l repetitions" - are you referring to k and l samples taken simultaneously (i.e. as a function of time), or "k and l" repetitions of the task (and if so, which time point are you looking at)?

"removing the differences in the starting position" → this is a limitation of the technique - the movements may be highly influenced by differences in starting position, this should be discussed later

"with respect to the coordination strategy objectified from the intial dataset" - does this mean that is the process is repeated but switching A and B different results are obtained?

page 7 - "on the opposite" → should be something like "in contrast"

- can you explain what is the significance of the sign and magnitude of the JcvPCA?

page 8 - "careful consideration" - are their some guidelines for this? how can this be avoided? filtering the data first?

- "if the movement times are similar" - the reason for using DTW or similar is not just because the movement times may be dissimilar, but rather because the relative amount of time for parts of the movement may be different and so should be normalized first

- such as DTW - give a reference to DTW. It would be good to note here also the use of "registration" (as is used in the functonal data analysis book) to solve this problem

"are plotted together" - isn't this Figure 2C rather than 2A?

"at each time step" (Fig 2B) - isn't this Figure 2D?

page 9 - JsvCRP A,B - isn't this definition problematic, because if for some of the time CRP B is bigger than A, and some of the time A is bigger than B (as in Figure 2E), won't the + and - cancel out? Should it be the absolute difference instead?

- also why 100 here? shouldn't this be more general (I assume this is for 100 time points?)

- Can you explain what is the meaning of 3.06 as the JsvCRP - is this a big or small number?

page 10 - what is the end effector here? did the wrist / fingers play a role?

Figure 4 - should joint position be joint angle (if it is in radians?)

page 11 - it would be helpful to include again here the intermediate steps in calculating jcvPCA (in the figure)

page 12 - it would be helpful to understand whether 2411 and 8272 are considered small or large numbers

page 16 - "for example, highly skilled players" why is it better to use this measure rather than just look at the "raw" joint angles or positions?

ergonomics - what sort of changes in these two numbers would be considered "good" or "bad" ?

rehab - again, what sort of changes do we expect to see?

References

reference 1 - missing article number (this journal does not use page numbers)

reference 7 - missing article number (this journal does not use page numbers)

reference 8 - missing a journal

reference 13 - missing details

reference 20 - journal name appears twice

reference 23 - missing article number (this journal does not use page numbers)

reference 25 - missing article number (this journal does not use page numbers)

reference 28 - missing article number (this journal does not use page numbers)

reference 32 - missing details

reference 33 - missing article number (this journal does not use page numbers)

reference 44 - missing article number

reference 50 - missing article number

reference 55 - missing article number

reference 59 - missing article number

reference 60 - missing article number

Figure 3 - Hand's projected should be hand's projection

6. PLOS authors have the option to publish the peer review history of their article (what does this mean?). If published, this will include your full peer review and any attached files.

Reviewer #1: No

Reviewer #2: No

---

## [Author Response · Author response to Decision Letter 1]

7 Jun 2024

All the reviewer comments have been answered in the rebuttal letter attached with the manuscript.

---

## [Decision Letter · Decision Letter 1]

15 Mar 2025

PONE-D-24-06791R1JcvPCA and JsvCRP : a set of metrics to evaluate changes in joint coordination strategiesPLOS ONE

Dear Dr. Dubois,

Thank you for submitting your manuscript to PLOS ONE. After careful consideration, we feel that it has merit but does not fully meet PLOS ONE’s publication criteria as it currently stands. Therefore, we invite you to submit a revised version of the manuscript that addresses the points raised during the review process.

We look forward to receiving your revised manuscript.

Kind regards,

Roberto Di Marco, PhD

Academic Editor

PLOS ONE

Reviewers' comments:

Reviewer's Responses to Questions

**Comments to the Author**

1. If the authors have adequately addressed your comments raised in a previous round of review and you feel that this manuscript is now acceptable for publication, you may indicate that here to bypass the “Comments to the Author” section, enter your conflict of interest statement in the “Confidential to Editor” section, and submit your "Accept" recommendation.

Reviewer #2: All comments have been addressed

Reviewer #3: (No Response)

2. Is the manuscript technically sound, and do the data support the conclusions?

Reviewer #2: Yes

Reviewer #3: Yes

3. Has the statistical analysis been performed appropriately and rigorously? 

Reviewer #2: Yes

Reviewer #3: N/A

4. Have the authors made all data underlying the findings in their manuscript fully available?

Reviewer #2: Yes

Reviewer #3: Yes

5. Is the manuscript presented in an intelligible fashion and written in standard English?

Reviewer #2: Yes

Reviewer #3: Yes

6. Review Comments to the Author

Reviewer #2: (No Response)

Reviewer #3: In this paper, the authors propose two indices based on PCA and CRP, respectively, to quantify joint coordination during movements. The aim of the study is interesting, and the paper does not present major technical or scientific issues. However, it is not well-structured, making it difficult to understand.

A scientific paper should include the following sections:

• Introduction

• Materials and Methods

• Results

• Discussion

• Conclusions

This paper contains numerous sections and subsections that do not strictly follow this structure. Therefore, I recommend extensive editing to improve clarity.

Starting with the Introduction section, the introduction should:

• Provide background that puts the manuscript into context and allows readers outside the field to understand the purpose and significance of the study

• Define the problem addressed and explain its importance

• Include a brief review of key literature

• Highlight any relevant controversies or disagreements in the field

• Conclude with a brief statement of the overall aim of the work and a comment on whether that aim was achieved

I suggest that the authors remove all subsections related to the introduction to ensure the key points outlined above are clear. Sections 1.2.1 and 1.2.2 could be summarized using references to papers that explain PCA and CRP.

Additionally, I recommend removing Section 2 (Notations) from the main text and turning it into an appendix.

Materials and Methods

It appears that Section 3 marks the beginning of the Materials and Methods section. This section requires reformatting as it is too confusing and does not allow readers to replicate the procedure proposed by the authors. I suggest structuring it as follows:

2 Materials and Methods

2.1 Joint contribution variations based on PCA reprojection (JcvPCA) and explanation of the calculation

2.2 Spatio-temporal synchronization between joints based on CRP (JsvCRP) and explanation of the calculation

2.3 Application to Experimental Data

Results

The results section is completely missing. The authors should separate the results from the methods section and provide appropriate commentary.

Discussion

The discussion is very clear and well-written. However, I question why the section is titled Discussion and Conclusions when a separate Conclusions section follows later. Ensure consistency in this regard.

Bibliography

Pay attention to the formatting of the references. The first citation in the text appears as number 54, but it should be 1. I suggest updating the references to follow the journal's style.

7. PLOS authors have the option to publish the peer review history of their article (what does this mean?). If published, this will include your full peer review and any attached files.

Reviewer #2: No

Reviewer #3: No

---

## [Author Response · Author response to Decision Letter 2]

7 Apr 2025

Dear Professor Di Marco and Reviewers,

We sincerely appreciate the time and effort you have dedicated to evaluating our manuscript, "JcvPCA and JsvCRP: A Set of Metrics to Evaluate Changes in Joint Coordination Strategies" (Manuscript ID: PONE-D-24-06791R1). We are grateful for the insightful comments, which have helped us enhance the quality and clarity of our work. Below, we provide detailed responses to each of the reviewers’ comments.

Reviewer 3 : In this paper, the authors propose two indices based on PCA and CRP, respectively, to quantify joint coordination during movements. The aim of the study is interesting, and the paper does not present major technical or scientific issues. However, it is not well-structured, making it difficult to understand.

A scientific paper should include the following sections:

• Introduction

• Materials and Methods

• Results

• Discussion

• Conclusions

This paper contains numerous sections and subsections that do not strictly follow this structure. Therefore, I recommend extensive editing to improve clarity.

Starting with the Introduction section, the introduction should:

• Provide background that puts the manuscript into context and allows readers outside the field to understand the purpose and significance of the study

• Define the problem addressed and explain its importance

• Include a brief review of key literature

• Highlight any relevant controversies or disagreements in the field

• Conclude with a brief statement of the overall aim of the work and a comment on whether that aim was achieved

I suggest that the authors remove all subsections related to the introduction to ensure the key points outlined above are clear. Sections 1.2.1 and 1.2.2 could be summarized using references to papers that explain PCA and CRP.

We greatly appreciate your feedback on the structure of the manuscript. We agree that a clear and logically organized structure is crucial for accessibility and comprehension. In response, we have undertaken significant revisions to improve the organization of the manuscript and ensure it aligns with conventional scientific formatting.

Introduction:

We recognize that the introduction section required substantial refinement. In response to your comment, we have completely rewritten the Introduction to streamline its presentation and ensure clarity. Specifically, we have removed the subsections in line with your recommendation, thereby eliminating unnecessary segmentation. Instead, we provide a concise narrative that integrates background context, the problem at hand, its significance, a brief review of key literature, and our study’s aim.

To improve readability and ease of understanding, we have replaced the first two figures with a single figure summarizing both the PCA and CRP metrics, simplifying these complex concepts without overwhelming the reader with excessive text. Additionally, we have added references to key papers explaining PCA and CRP, ensuring that these important concepts are sufficiently contextualized for readers outside the field.

Additionally, I recommend removing Section 2 (Notations) from the main text and turning it into an appendix.

Regarding your suggestion to move the Notations section to the appendix, we fully agree and have implemented this change. The Notations section has now been moved to supplementary material, ensuring that the main text remains focused and streamlined. However, to preserve essential clarity, we have retained key details, such as the recommended number of principal components (PCs) to retain, which has been relocated to the beginning of Section 2.1.

Materials and Methods

It appears that Section 3 marks the beginning of the Materials and Methods section. This section requires reformatting as it is too confusing and does not allow readers to replicate the procedure proposed by the authors. I suggest structuring it as follows:

2 Materials and Methods

2.1 Joint contribution variations based on PCA reprojection (JcvPCA) and explanation of the calculation

2.2 Spatio-temporal synchronization between joints based on CRP (JsvCRP) and explanation of the calculation

2.3 Application to Experimental Data

We truly appreciate this suggestion, which we believe strengthens the clarity and usability of our work. As a result, we have thoroughly restructured the Materials and Methods section to adhere to a clearer, more coherent framework. The revised section is now organized as follows:

• Mathematical Framework for Joint Contribution Analysis Using PCA Reprojection (JcvPCA) (Page 3)

• Mathematical Framework for Spatio-Temporal Joint Synchronization Using CRP (JsvCRP) (Page 5)

• Data collection for validation of joint coordination metrics (Page 7): This section is now subdivided to separately describe the generation of both simulated and experimental datasets. The description of the simulated dataset, which was originally part of the Results section, has now been appropriately moved here.

• Data processing and analysis using joint coordination metrics (Page 8): This section describes the application of the metrics to the generated datasets.

This restructuring ensures that each step in our methodology is clearly defined, making it easier for readers to follow the approach and replicate the analysis.

Results

The results section is completely missing. The authors should separate the results from the methods section and provide appropriate commentary.

We acknowledge and regret this oversight, and we are grateful for your constructive suggestion. In response, we have now added a dedicated Results section (Section 3, Page 9) to clearly differentiate the results from the methods. The results are divided into two subsections:

1. Validation using the simulated dataset

2. Validation using the experimental dataset

This division ensures that the experimental findings are presented in a clear and coherent manner, separated from the detailed methodological explanations. An additional paragraph on the way to use both JcvPCA and JsvCRP together to analyse inter-joint coordination data has been added at the end of the Results section ( 3.3 Interpretation of JsvCRP and JcvPCA together to characterize coordination schemes)

Discussion

The discussion is very clear and well-written. However, I question why the section is titled Discussion and Conclusions when a separate Conclusions section follows later. Ensure consistency in this regard.

We appreciate your attention to detail and agree that consistency in section titles is important for clarity. As per your suggestion, we have modified the title of the “Discussion and Conclusions” section to simply Discussion, ensuring alignment with common academic practices. The conclusions are now appropriately placed in a separate Conclusions section at the end of the manuscript, ensuring clarity and a clean distinction between the discussion of results and the study's final remarks.

Bibliography

Pay attention to the formatting of the references. The first citation in the text appears as number 54, but it should be 1. I suggest updating the references to follow the journal's style.

We appreciate your observation regarding the formatting of the references. As recommended, we have carefully reviewed and updated the reference list, ensuring that citations are now numbered sequentially according to their first mention in the text. Additionally, we have formatted the bibliography to comply with the journal’s style guidelines.

We have added a paragraph in the Discussion section to clearly emphasize that the JcvPCA and JsvCRP metrics evaluate variations in joint coordination strategies, and we specifically clarify that JcvPCA is not designed to compare fundamentally different coordination strategies. This addition aims to ensure that readers fully understand the context and limitations of the proposed metrics.

We trust that these revisions adequately address your concerns. We are grateful for your thorough review, which has greatly contributed to improving the manuscript. Please let us know if any further modifications or clarifications are required.

Thank you once again for your time and thoughtful feedback.

Sincerely,

Océane DUBOIS

---

## [Decision Letter · Decision Letter 2]

8 May 2025

PONE-D-24-06791R2JcvPCA and JsvCRP : a set of metrics to evaluate changes in joint coordination strategiesPLOS ONE

Dear Dr. Dubois,

Thank you for submitting your manuscript to PLOS ONE. After careful consideration, we feel that it has merit but does not fully meet PLOS ONE’s publication criteria as it currently stands. Therefore, we invite you to submit a revised version of the manuscript that addresses the points raised during the review process.

We look forward to receiving your revised manuscript.

Kind regards,

Roberto Di Marco, PhD

Academic Editor

PLOS ONE

Journal Requirements:

Reviewers' comments:

Reviewer's Responses to Questions

**Comments to the Author**

1. If the authors have adequately addressed your comments raised in a previous round of review and you feel that this manuscript is now acceptable for publication, you may indicate that here to bypass the “Comments to the Author” section, enter your conflict of interest statement in the “Confidential to Editor” section, and submit your "Accept" recommendation.

Reviewer #3: (No Response)

2. Is the manuscript technically sound, and do the data support the conclusions?

Reviewer #3: Yes

3. Has the statistical analysis been performed appropriately and rigorously? 

Reviewer #3: N/A

4. Have the authors made all data underlying the findings in their manuscript fully available?

Reviewer #3: Yes

5. Is the manuscript presented in an intelligible fashion and written in standard English?

Reviewer #3: Yes

6. Review Comments to the Author

Reviewer #3: I would like to thank the authors for their thorough review and for responding to each of my comments. In my opinion, the work has significantly improved and is worthy of being published. I have a curiosity, and I would like to point out a couple of typos.

• Just out of curiosity, I wanted to know what the rationale was for choosing the number of PCs as p+1 instead of, for example, the number of PCs that explain at least 95% of the variance?

• At the beginning of page 7, there are typos in the sentence 'Experimental protocol The participant was asked to reach each of the 3 targets 5 times using different coordination strategies.' There is a lack of punctuation, and some words are in bold.

• The numbering of the figures should follow the order of appearance in the text, but figure 4 is mentioned first, then figure 3, and then figure 2. I recommend rearranging them

7. PLOS authors have the option to publish the peer review history of their article (what does this mean?). If published, this will include your full peer review and any attached files.

Reviewer #3: No

---

## [Author Response · Author response to Decision Letter 3]

14 May 2025

Dear Editor,

We thank you and the reviewers for the time and effort dedicated to evaluating our manuscript. We greatly appreciate Reviewer’s thoughtful and constructive feedback, and we have carefully addressed all points raised. Below, we provide a detailed response to the reviewer’s comments and the editorial request regarding the references.

Editorial Request: References

We have carefully reviewed our reference list to ensure it is complete and correct. We have added all DOIs when possible. No retracted papers remain in the reference list.

Reviewer #3 Comments and Author Responses

Comment 1: "Just out of curiosity, I wanted to know what the rationale was for choosing the number of PCs as p+1 instead of, for example, the number of PCs that explain at least 95% of the variance?"

Response:

We thank the reviewer for this insightful question. In most cases, selecting the first p+1 principal components captures at least 95% of the variance, aligning well with common practices. However, our rationale for this choice stems from task-specific considerations. We framed the analysis in terms of the degrees of freedom necessary to perform the task rather than relying solely on variance explanation. This approach helps avoid overfitting or including extraneous components, especially in instances where participants employ atypical strategies that may introduce noise or irrelevant variability into the data.

Comment 2: "At the beginning of page 7, there are typos in the sentence 'Experimental protocol The participant was asked to reach each of the 3 targets 5 times using different coordination strategies.' There is a lack of punctuation, and some words are in bold."

Response:

We appreciate the reviewer pointing this out. The formatting issues and punctuation errors in this sentence have been corrected. The revised sentence now reads:

“Experimental protocol : The participant was asked to reach each of the three targets five times using different coordination strategies.”

Comment 3: "The numbering of the figures should follow the order of appearance in the text, but figure 4 is mentioned first, then figure 3, and then figure 2. I recommend rearranging them."

Response:

Thank you for highlighting this oversight. We have revised the manuscript to ensure that figures are introduced in numerical order. The figure references have been updated accordingly to reflect the proper order of appearance.

We hope that the changes made fully address the comments and suggestions provided. We are grateful for the opportunity to revise our manuscript and for the constructive feedback that has helped us improve the clarity and rigor of our work.

Sincerely,

Océane Dubois

---

## [Decision Letter · Decision Letter 3]

21 May 2025

JcvPCA and JsvCRP : a set of metrics to evaluate changes in joint coordination strategies

PONE-D-24-06791R3

Dear Dr. Dubois,

We’re pleased to inform you that your manuscript has been judged scientifically suitable for publication and will be formally accepted for publication once it meets all outstanding technical requirements.

Kind regards,

Roberto Di Marco, PhD

Academic Editor

PLOS ONE

Additional Editor Comments (optional):

Reviewers' comments:

Reviewer's Responses to Questions

**Comments to the Author**

1. If the authors have adequately addressed your comments raised in a previous round of review and you feel that this manuscript is now acceptable for publication, you may indicate that here to bypass the “Comments to the Author” section, enter your conflict of interest statement in the “Confidential to Editor” section, and submit your "Accept" recommendation.

Reviewer #3: All comments have been addressed

2. Is the manuscript technically sound, and do the data support the conclusions?

Reviewer #3: (No Response)

3. Has the statistical analysis been performed appropriately and rigorously? 

Reviewer #3: (No Response)

4. Have the authors made all data underlying the findings in their manuscript fully available?

Reviewer #3: (No Response)

5. Is the manuscript presented in an intelligible fashion and written in standard English?

Reviewer #3: (No Response)

6. Review Comments to the Author

Reviewer #3: (No Response)

7. PLOS authors have the option to publish the peer review history of their article (what does this mean?). If published, this will include your full peer review and any attached files.

Reviewer #3: **Yes: **Emilia Scalona

---

## [Editor Report · Acceptance letter]

PONE-D-24-06791R3

PLOS ONE

Dear Dr. Dubois,

I'm pleased to inform you that your manuscript has been deemed suitable for publication in PLOS ONE. Congratulations! Your manuscript is now being handed over to our production team.

Kind regards,

on behalf of

Dr. Roberto Di Marco

Academic Editor

PLOS ONE